# ELICITING ATTRIBUTIONS FROM LLMS WITH MINIMAL SUPERVISION

## ABSTRACT

Large Language Models (LLMs) have quickly become popular tools for recalling information; however, the specific mechanisms they utilize to encode and store vast information within their parameters is not well understood. As a step towards improving interpretability and reliability of LLMs, we develop `AttributeLLaMA`, which involves fine-tuning LLaMA models to enable them to attribute their responses. By utilizing a mere 100 expert-annotated attribution examples, we are able to achieve this capability. Our experimental studies demonstrate that these attributions significantly improve the performance of a strong LLM on downstream tasks such as MMLU and StrategyQA by more than 3.5% and 4.0%, respectively. Furthermore, our analyses on the attributions obtained from `AttributeLLaMA` reveal the remarkable memorization capabilities of LLMs.

## 1 INTRODUCTION

Large Language Models (LLMs) have quickly become popular tools for information-seeking contexts. However, determining the underlying factors responsible for the content generated by LLMs is often challenging. This challenge arises from the fact that LLMs undergo pre-training on large datasets, and the specifics of how they store and encode this vast information within their parameters are not well understood. Thus, exploring methods that enable LLMs to attribute their own responses would be highly beneficial.

The attribution process involves identifying the source information that the model utilizes to generate its content. A recent study (Bohnet et al., 2022) investigated methods to enable LLMs to attribute information in question answering tasks. In addition to retrieving and attributing information, the study explores the feasibility of employing post-hoc attribution for answers generated by LLMs. Another approach uses end-to-end modeling to generate a Wikipedia URL as an attribution for the generated answer. However, these prior studies have not addressed the issue of enabling the model to generate the actual attribution itself. Our work focuses on precisely this aspect.

In this work, we present `AttributeLLaMA`, an extension of LLaMA models (Touvron et al., 2023), finetuned on 100 expert-annotated attribution examples. Our approach aligns with previous research (Zhou et al., 2023), demonstrating the feasibility of modeling LLMs with high-quality but minimal data. One of the primary obstacles in attribution lies in determining how to measure it effectively. To address this, we introduce two types of attribution evaluations. The first type focuses on assessing whether attributions enhance the performance of downstream tasks. The second type involves measuring attribution quality through various ablations and analyses, including human evaluations.

To evaluate the impact of attributions on downstream tasks, we conduct an extensive analysis across a diverse range of tasks. We examine tasks that demand substantial world knowledge and problem-solving skills, such as MMLU (Hendrycks et al., 2021). We also study knowledge-intensive question answering tasks such as Natural Questions (Kwiatkowski et al., 2019), TriviaQA (Joshi et al., 2017), and WebQ (Berant et al., 2013). Additionally, we explore reasoning-related tasks, including ARC Easy/Challenge (Clark et al., 2018), OpenBookQA Mihaylov et al. (2018), and StrategyQA (Geva et al., 2021). Our findings demonstrate that LLaMA Touvron et al. (2023) models improve performance of these diverse tasks significantly when the relevant attributions are provided as additional context. On MMLU task the LLaMA models in a zero-shot setting improve by more than 3.5%

with the help of attributions. Similarly, on StrategyQA, attributions help LLaMA models improve by more than 4.0%. Further, we measure the quality of attributions from our `AttributeLLaMA` model using lexical and entity overlapping approaches w.r.t. relevant source documents and observe that `AttributeLLaMA` has remarkable memorization of Wikipedia documents, which are known to be in the training corpora of LLaMA.

In summary, our work makes the following contributions. Firstly, we introduce `AttributeLLaMA`, a model capable of providing attribution information pertaining to the generated response for a given query or prompt. Remarkably, we achieve this by using a mere 100 expert-annotated attribution examples. Furthermore, we propose and demonstrate various evaluation methods to assess the performance of these attribution models.

## 2 RELATED WORK

### 2.1 ATTRIBUTION

The task of attribution involves identifying the source information that the model utilizes to generate its response. The most trivial form of attribution is post-hoc attribution (Bohnet et al., 2022), where the model first generates a response for a given prompt/question, and later a query is formed with both prompt/question and response to retrieve relevant documents for attribution. Gao et al. (2022) further propose to post-edit the response mainly attributable to the retrieved documents. Another variant of attribution is using a differential retrieval where a neural network generates both the answer and a pointer to the attribution corpus in the form of document identifiers (Tay et al., 2022; Wang et al., 2022; Bohnet et al., 2022). Our work focuses on enabling the LLMs to generate their attribution using their parametric knowledge.

### 2.2 INSTRUCTION TUNING

Significant advancements have been made in various research domains by large language models, owing to their remarkable in-context learning capabilities (Brown et al., 2020), emergent behavior (Wei et al., 2022a), and chain-of-thought reasoning skills (Wei et al., 2022b). In recent times, instruction tuning and reinforcement learning with human feedback (RLHF) have gained popularity as effective methods to enhance the ability of LLMs to effectively follow instructions Ouyang et al. (2022); Wei et al. (2021); Chung et al. (2022); Iyer et al. (2022). In a typical instruction tuning scenario, LLMs are fine-tuned on a diverse range of tasks, including both traditional natural language processing (NLP) tasks and more general and practical tasks like composing poems or emails Ouyang et al. (2022). Moreover, there have been recent proposals for more efficient instruction-tuning methods, including optimizations to reduce computational overhead during fine-tuning Hu et al. (2021) and techniques for utilizing less instruction-following data Zhou et al. (2023). Notably, studies have demonstrated that high-quality instruction following models can be developed with as few as 1,000 examples of high-quality instruction data Zhou et al. (2023). Inspired by this prior research, we developed our `AttributeLLaMA` model by employing instruction fine-tuning of LLaMA models to generate attributions for any given question. Notably, we achieve this outcome with minimal data, requiring only 100 examples.

### 2.3 QUESTION ANSWERING TASKS

In this work, our attribution methods are primarily focused on the question-answering setting. Question answering is one of the most popular tasks in NLP, with a wide variety of settings that can be broadly categorized into closed-domain question answering, which involves reading comprehension tasks like SQuAD (Rajpurkar et al., 2016), and open-domain question answering, which involves acquiring information from extensive knowledge sources such as the web. SQuAD has emerged as a widely adopted and influential benchmark for closed question-answering, stimulating the development of other reading comprehension datasets (Choi et al., 2018; Reddy et al., 2019). In the open-domain QA, Natural Questions (Kwiatkowski et al., 2019) provides a large-scale dataset based on information-seeking queries from the Google search engine. Subsequently, several additional information-seeking works have been published, such as WebQ (Berant et al., 2013).

In this study, our particular emphasis lies on open-domain question answering and its attribution. However, we specifically concentrate on generating attributions from the inherent knowledge of the language model itself, utilizing its parametric knowledge rather than attributing answers to externally retrieved documents.

## 3   APPROACH

To enable a model to generate the source information, which is the basis for arriving at an answer, we leverage the impressive generation capabilities of modern LLMs. Prior research framed the objective as generating a reference to the source (Menick et al., 2022), using either full supervision or in-context learning methods (Bohnet et al., 2022). In our work, we explore directions to allow the model to generate the source passage, leveraging its own memorization capabilities (Carlini et al., 2021).

We operate in the classic QA setup. Given a prompt $p$ with question $q$, the model's objective is to generate the answer $a$. In our setup, we additionally prompt the model to generate an evidence or *attribution* passage $e$ along with the answer. We typically ensure the model generates the passage first and then the answer, following the Chain-of-Thought prompting literature Wei et al. (2022b). The input prompt can be as follows:

> *Answer the following question by first generating an evidence and using it to answer. Question:* $(q)$ *Evidence:* $(e)$ *Answer:* $(a)$

In order to produce the evidence $e$ followed by the answer $a$, the model requires *alignment*. This alignment can be achieved through an in-context learning setup, where the model is provided $k$ *exemplars* ($k$-shot). However, a major drawback of in-context learning approaches is that the evidences of the $k$ exemplars can be arbitrarily long, thus limiting the number of examples to fit in the context. Additionally, this also increases the computational cost of model inference.

Another popular approach to model alignment is training through supervision. Given a high-quality question-answer-evidence dataset, an LLM can be fine-tuned to elicit responses for unseen questions. However, the biggest roadblock to such research is the lack of high-quality data. The Natural Questions dataset (Kwiatkowski et al., 2019) is one of the closest matches, providing each question with a short and long answer. The long answer can be used as a proxy for attribution, as it is derived from one or more short spans from the Wikipedia passage that contains the actual answer. However, supervision can be tricky for alignment. While more supervision could be beneficial, it might also impose the finetuning dataset knowledge on the LLM, making the evidences generated less equivalent to the pre-training corpus (studied in Sec. 5). Therefore, we explore the route of the *minimal supervision* (Zhou et al., 2023) paradigm in this work, where a small set of high-quality attributions are annotated by experts. This allows us to create a more comprehensive and accurate dataset of attributions that could be used to train attribution models.

In this work, we specifically investigate methods to elicit attributions from the recently released LLaMA (Touvron et al., 2023) family of models. We coin the model aligned to provide attributions as `AttributeLLaMA`, and investigate various methods for training the model and evaluating the quality of the generated attributions.

### 3.1   MINIMAL SUPERVISION WITH HIGH QUALITY DATA

In our initial analysis, we found that the long answers in the Natural Questions dataset (NQ) were too short and limited to be considered as attribution. Specifically, the NQ finetuned LLaMA models tend to overfit on the NQ distribution, leading to poor generalization (studied in Sec. 5). We collected high-quality question-answer-attribution triplets inspired by recent findings of Zhou et al. (2023) on *minimal supervision*. Specifically, we extracted a subset of open-domain QA task examples from the Dolly dataset,[1] which contains question-answer pairs, where the question can be answered using world knowledge. Then, we had two expert annotators construct attributions to these pairs by referring to information from Wikipedia. The general guideline provided to the annotators was to refer to the exact Wikipedia passage as much as possible and remove unwanted information such as

---

[1] https://huggingface.co/datasets/databricks/databricks-dolly-15k

excessive dates, pronunciations, and abbreviations that are not directly related to the question. We further filtered down the annotations by cross-examination to ensure that the attribution alone can be used as the context to answer a given question. This expert annotation process allowed us to collect high-quality and vetted 100 examples, which we used to train `AttributeLLaMA`. Table 7 of the Appendix shows a few examples from this annotation.

## 3.2 TRAINING DETAILS

`AttributeLLaMA` is based on the recently published LLaMA (Touvron et al., 2023) model family, with minimal supervision from only 100 expert-annotated attribution examples. We use two variants: 7B and 65B models as the base for the `AttributeLLaMA`. Because we had limited data, we used a very small batch size and learning rate to allow for more steps. We used Adam optimizer with the initial learning rate of $1 \times 10^{-6}$ and a warm-up of 10 steps. We used a batch size of 8 and a maximum number of optimization steps of 100 for both the 7B and 65B models. The objective function is only optimized on the loss from the target tokens, i.e., instruction/prompt is not part of the loss optimization.

## 4 EXPERIMENTS & RESULTS

This section critically evaluates the quality of the attributions generated by `AttributeLLaMA`. Examples of the generation are provided in Table 6. Evaluating the factuality of an LLM-generated passage is a complex problem and is an active research area (Yue et al., 2023; Min et al., 2023; Liu et al., 2023; Manakul et al., 2023; Fu et al., 2023). Bohnet et al. (2022) use a variant of Natural Language Inference based automatic attribution evaluation (Rashkin et al., 2021) model, which rates the factuality of the attribution given a ground truth reference or a retrieved document based on the passage. Others (Manakul et al., 2023; Yue et al., 2023; Min et al., 2023) devise methods to automatically rate the factuality of a generated attribution using an LLM. It is debatable which method is better - retrieval or non-retrieval LLM-based methods. Thus, in our work, we evaluate the generated attributions in light of their *usefulness* in a given downstream task. Following the setup from (Yu et al., 2023), we evaluate the attributions in a two-step process: in the first step, we generate an attribution passage given a question-answer pair from a downstream task, and in the second step, we use the attribution as the context for the model to answer the question. Concretely, given the attribution generated by `AttributeLLaMA` (using a prompt), we construct an input prompt such as:

> Use the following context to answer: ($e$) Question: ($q$) Answer:

In case of $k$-shot tasks, we use the same exemplars as the baseline task and insert the context before the query:

> {EXEMPLARS} Use the following context to answer: ($e$) Question: ($q$) Answer:

We then evaluate the change in model performance with this additional context. If the attribution is *relevant* to the given question, then the model should benefit from having it in the context, similar to the progress in Chain-of-Thought prompting literature. We evaluate popular knowledge-intensive tasks such as Natural Questions, WebQ, TriviaQA from the GenRead benchmark (Yu et al., 2023), MMLU (Hendrycks et al., 2021) and reasoning tasks such as StrategyQA (Geva et al., 2021), ARC-Easy, ARC-Challenge (Clark et al., 2018) and OpenBookQA (Mihaylov et al., 2018).

## 4.1 MMLU BENCHMARK

MMLU Hendrycks et al. (2021) is a popular benchmark for testing models on knowledge-intensive multi-choice questions. The benchmark has examples spanning the subjects of humanities, social science, and other areas making up a total of 57 categories. To evaluate the usefulness of our `AttributeLLaMA` models, we derive attributions for each of the test examples of the MMLU benchmark and use it as the context in both 0-shot and 5-shot settings. Table 1 presents the results comparing LLaMA models with and without attributions. For each model size (7B and 65B), we provide the attributions from our `AttributeLLaMA` 7B and 65B models as context.

Table 1: Performance comparison of LLaMA models on MMLU benchmark with and without attributions from `AttributeLLaMA` models.

| Model Name | Accuracy (0-shot) | Accuracy (5-shot) |
|---|---|---|
| LLaMA-7B | 28.6 | 35.1 |
| + `AttributeLLaMA`-7B | 32.1 | 38.6 |
| + `AttributeLLaMA`-65B | **34.0** | **41.2** |
| LLaMA-65B | 54.8 | 63.4 |
| + `AttributeLLaMA`-7B | 54.1 | 60.7 |
| + `AttributeLLaMA`-65B | **58.7** | **64.0** |

Table 2: Performance comparison of LLaMA models on open-domain question answering tasks with and without attributions from `AttributeLLaMA` models. EM denotes exact match score and RM denotes relaxed match score.

| Model Name | NQ | | TriviaQA | | WebQ | |
|---|---|---|---|---|---|---|
| | EM | RM | EM | RM | EM | RM |
| InstructGPT-175B | 20.9 | - | 56.7 | - | 19.0 | - |
| GenRead-175B | 28.0 | - | 59.0 | - | 24.6 | - |
| LLaMA-7B (4-shot) | 21.8 | 25.9 | 56.6 | 59.5 | 30.0 | 39.8 |
| + `AttributeLLaMA`-7B | 23.3 | 28.1 | 52.7 | 56.2 | 28.0 | 38.2 |
| + `AttributeLLaMA`-65B | **30.8** | **35.6** | **62.3** | **65.8** | **30.0** | **40.8** |
| LLaMA-65B (4-shot) | 36.6 | 40.6 | 73.5 | 76.5 | 39.9 | 45.1 |
| + `AttributeLLaMA`-7B | 30.2 | 34.7 | 63.7 | 66.9 | 33.5 | 40.1 |
| + `AttributeLLaMA`-65B | 35.3 | 40.1 | 68.8 | 72.3 | 34.5 | 42.8 |

We observe that in the zero-shot setting, the attributions from both 7B and 65B `AttributeLLaMA` models significantly improve the base model's performance. For LLaMA-7B, the improvements are 3.5% and 5.4% with `AttributeLLaMA` 7B and 65B models, respectively. For LLaMA-65B, `AttributeLLaMA`-7B hurts the performance, whereas `AttributeLLaMA`-65B still improves the performance by 3.9%. This suggests that attributions from bigger model sizes can help the smaller models, but not the opposite.

In the 5-shot setting, we only provide the attributions as context for the test example, while the few-shot exemplars remain the same. For the LLaMA-7B model, we again observe improvements of 3.5% and 6.1% with the `AttributeLLaMA` 7B and 65B models, respectively. These improvements are in-line with that of the 0-shot setting. We also observe improvements for the LLaMA-65B model in the 5-shot setting with attributions, but these improvements are minor for the 0-shot setting. We hypothesize that this is because the in-context setting might become powerful at a large scale and possibly give less weight to attributions.

## 4.2 OPEN-DOMAIN QUESTION ANSWERING TASKS

We also evaluate the quality of our `AttributeLLaMA` on the GenRead benchmark (Yu et al., 2023), specifically, we evaluate on the following open-domain QA datasets: Natural Questions (NQ) (Kwiatkowski et al., 2019), TriviaQA (Joshi et al., 2017), and WebQ (Berant et al., 2013). We use the exact match (EM) score, where the normalized form of the predicted answer is an exact match w.r.t. one of the ground-truth answers' normalized forms. We also use a relaxed match (RM) score to check if the normalized form of the predicted answer is a sub-string match w.r.t. one of the ground-truth answers' normalized forms.

Table 2 presents the results on these QA tasks compared against the previous state-of-the-art results. First, we note that the LLaMA-7B baseline model with a 4-shot setting is already strong and beats the instruction-tuned GPT-175B model in 2 out of 3 tasks. Upon adding the attributions from our `AttributeLLaMA` to LLaMA-7B, we observe significant improvements in beating GenRead-

Table 3: Performance comparison of LLaMA models on zero-shot knowledge-intensive tasks with and without attributions from `AttributeLLaMA` models.

| Model Name | StrategyQA (0-shot) | StrategyQA (5-shot) |
|---|---|---|
| LLaMA-7B | 46.3 | 47.9 |
| + `AttributeLLaMA`-7B | 47.7 | 51.3 |
| + `AttributeLLaMA`-65B | **48.0** | **53.6** |
| LLaMA-65B | 50.7 | 62.8 |
| + `AttributeLLaMA`-7B | 52.1 | 65.0 |
| + `AttributeLLaMA`-65B | **52.6** | **67.1** |

Table 4: Performance comparison of LLaMA models on zero-shot reasoning tasks with and without attributions from `AttributeLLaMA` models.

| Model Name | ARC-Easy | ARC-Challenge | OpenBookQA |
|---|---|---|---|
| LLaMA-7B | 76.2 | 42.7 | 31.2 |
| + `AttributeLLaMA`-7B | 74.3 | 43.3 | 36.2 |
| + `AttributeLLaMA`-65B | **78.2** | **44.6** | **38.4** |
| LLaMA-65B | **81.0** | **52.8** | 37.0 |
| + `AttributeLLaMA`-7B | 77.1 | 48.2 | **40.0** |
| + `AttributeLLaMA`-65B | 80.3 | 49.2 | 39.4 |

175B approach in 1 out of 3 tasks with attributions from `AttributeLLaMA`-7B, and in all three tasks with attributions from `AttributeLLaMA`-65B. We also evaluate the LLaMA-65B model, where we observe that attributions hurt performance. We hypothesize that as the model scale increases, the few-shot setting becomes more powerful and possibly gives less weight to attributions.

### 4.3 REASONING TASKS

Attribution is a general concept that extends beyond information-seeking question answering tasks. For example, it can be applied to tasks that require reasoning to solve, given that reasoning involves drawing inferences from facts, which can be attributed. Therefore, in this section, we study the usefulness of attributions on different reasoning tasks. Note that our attribution models are developed focusing on question-answering tasks. Hence their generalization to all types of reasoning tasks may be limited. Nevertheless, we demonstrate that our attribution models can help improve performance on reasoning tasks, as evidenced in Table 4 and Table 3.

Table 3 shows the effectiveness of attributions on the StrategyQA task Geva et al. (2021), which is a question-answering benchmark with required reasoning steps implicit in the question. We consistently observe improvements when attributing this task to both LLaMA 7B and 65B models. In the 0-shot setting, LLaMA-7B improves by 1.4% and 1.7% with `AttributeLLaMA` 7B and 65B, respectively. LLaMA-65B model improves by 1.4% and 1.9% with `AttributeLLaMA` 7B and 65B, respectively. Surprisingly, we find that the improvements are much larger in the 5-shot setting than in the 0-shot setting, where LLaMA-7B improves by 3.4% and 5.7%, and LLaMA-65B improves by 2.2% and 4.3% with `AttributeLLaMA` 7B and 65B, respectively. This trend is the opposite of what we observe on the MMLU benchmark.

In Table 4, we study how attributions affect performance on the following three reasoning tasks in a zero-shot setting: ARC-Easy (Clark et al., 2018), ARC-Challenge (Clark et al., 2018), and OpenBookQA (Mihaylov et al., 2018). For LLaMA-7B, attributions from `AttributeLLaMA`-7B help in 2 out of 3 tasks, whereas `AttributeLLaMA`-65B help in all three tasks. For LLaMA-65B, we observe that attributions only improve the performance on the OpenBookQA task. Note that attributions might not always help improve the reasoning tasks, but here we showed it could be possible to improve some of them.

Table 5: Overlap scores between the attributions generated on NQ and WebQ datasets and their corresponding retrieved Wikipedia documents, using `AttributeLLaMA`-65B and its ablations - Instruct and Supervised. Numbers in the (.) denote standard deviations.

| | Dataset | ROUGE-2 | | | ROUGE-L | | |
|---|---|---|---|---|---|---|---|
| | | Precision | Recall | F-score | Precision | Recall | F-score |
| Instruct | NQ | 0.24 (0.21) | 0.27 (0.19) | 0.24 (0.18) | 0.31 (0.22) | 0.35 (0.20) | 0.30 (0.18) |
| | WebQ | 0.06 (0.16) | 0.07 (0.16) | 0.06 (0.15) | 0.08 (0.18) | 0.09 (0.20) | 0.08 (0.17) |
| Supervised | NQ | 0.34 (0.28) | 0.48 (0.32) | 0.37 (0.29) | 0.40 (0.27) | 0.58 (0.29) | 0.44 (0.27) |
| | WebQ | 0.28 (0.23) | 0.44 (0.29) | 0.31 (0.24) | 0.34 (0.23) | 0.55 (0.26) | 0.39 (0.23) |
| `AttributeLLaMA` | NQ | 0.31 (0.24) | 0.39 (0.24) | 0.33 (0.23) | 0.37 (0.23) | 0.49 (0.22) | 0.40 (0.21) |
| | WebQ | 0.36 (0.23) | 0.45 (0.24) | 0.38 (0.22) | 0.42 (0.23) | 0.54 (0.22) | 0.45 (0.21) |

## 5 ANALYSIS OF ATTRIBUTION QUALITY

In this section, we analyze the attribution quality of `AttributeLLaMA`. Since `AttributeLLaMA` is trained with manual annotations from Wikipedia, the model is expected to elicit attributions in the same style and format. However, does the model output Wikipedia documents *verbatim*? In our preliminary experiments, it quickly became clear `AttributeLLaMA` is displaying remarkable memorization of the Wikipedia documents, which are known to be in the training corpora (Touvron et al., 2023). Additionally, our analyses suggest that attributions are prone to hallucinations. Therefore this section analyzes the quality of attributions generated by `AttributeLLaMA` from the perspective of memorization and hallucination.

### 5.1 ABLATIONS

We study attributions from various methods, including our `AttributeLLaMA`, an instruct model, and a fully-supervised model. We briefly describe each of these methods below.

**Instruct Model.** Several approaches to instruction-tuned LLMs have recently resulted in impressive general-purpose chat agents, primarily led by the success of ChatGPT models from OpenAI. Among various replications of the success of ChatGPT, the recently released Dolly dataset (also known as `databricks-dolly-15k` dataset) is increasingly used to finetune publicly accessible models such as LLaMA. We, therefore, investigate whether such a high-quality general-purpose dataset can also be used to elicit attributions from the base model. The Dolly dataset contains a subset of closed-domain QA, where the example is supported by a context that should be used to answer the question. We changed the format and prompts for this subset of question-answer-context triplets, and repurposed them to be an attribution of the question-answer pairs, resulting in 200 modified examples within the full 15k data.

**Fully-Supervised Model.** We finetuned LLaMA models on a large corpus of question-answer-attribution triplets from the Natural Questions (NQ) dataset (Kwiatkowski et al., 2019). The long answers in the NQ dataset are used as a proxy for attribution. We filter the dataset based on the length of the long answers, and end up using 87,000 training examples.

### 5.2 MEMORIZATION

Memorization exhibited by LLM's is not a new finding - over the last couple of years, many works have provided conclusive evidence of the same (Biderman et al., 2023; Carlini et al., 2021; 2022). Typically memorization is defined in the notion of $k$-elicitable (Biderman et al., 2023; Carlini et al., 2022) - the capability of the model to complete the generation of a string given $k$ previous tokens, with the assumption that the string exists in the training data. In the attribution QA setup of `AttributeLLaMA` the notion of $k$ previous tokens is difficult to define as we devise the instructions that we use to train and infer. While these papers start from curated evaluation data from a known training corpus, we approach the analysis differently. Given a model-generated attribution, we query a Wikipedia dump (provided by Bohnet et al. (2022)) to get the closest match. Concretely,

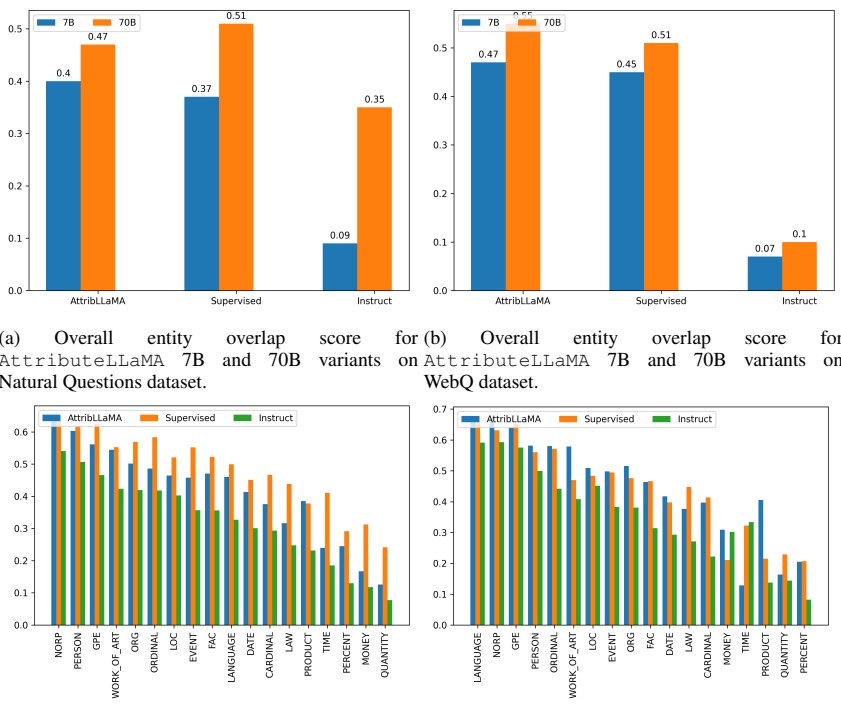

(a) Overall entity overlap score for `AttributeLLaMA` 7B and 70B variants on Natural Questions dataset.

(b) Overall entity overlap score for `AttributeLLaMA` 7B and 70B variants on WebQ dataset.

(c) Entity overlap broken down by Named Entity Types in Natural Questions dataset.

(d) Entity overlap broken down by Named Entity Types in WebQ dataset.

Figure 1: Named entity precision scores on `AttributeLLaMA` variants with Top-1 retrieved Wikipedia documents.

we use BM25 to perform retrieval from Wikipedia using each generated attribution as a query, leveraging the Pyserini (Lin et al., 2021) toolkit.

The lexical overlap scores shown in Table 5 highlight the amount of memorization the attribution methods have w.r.t. the Wikipedia text (see also example in Figure 2). The high recall of the ROUGE-L (Lin, 2004) scores indicates that the generated attributions have a high degree of verbatim overlap with the retrieved passage (Table 6). Among the variants, `AttributeLLaMA` achieves the best ROUGE scores on the WebQ dataset, while the fully-supervised model achieves the best scores on the NQ dataset. The instruct models have the lowest overlap, especially on the WebQ dataset. Since the fully-supervised model is trained on the Wikipedia-focused NQ distribution, its attributions will naturally have higher overlap when tested with NQ. This is evident from the fact that `AttributeLLaMA` has better overlap scores on out-of-domain WebQ. Overall, `AttributeLLaMA` has the highest overlap on average across both datasets. This shows that with careful guidance, the models can be primed to generate attributions from their own learned corpora, highlighting their remarkable memorization ability.

## 5.3 HALLUCINATIONS

We also intend to quantify *hallucination* for the generated attributions. One way of estimating hallucination is to compute named entity overlap precision of the generated attributions with respect to the top-$k$ retrieved Wikipedia documents.[2] Figure 1a and Figure 1b present the entity precision overlap scores comparing different attribution methods across 7B and 65B model parameter sizes. We observe that `AttributeLLaMA` and fully-supervised models have better scores on NQ and WebQ datasets, respectively. Here also, we believe that fully-supervised model is doing better on NQ since it was trained on the same dataset. We also look at the overlap precision with respect to various named entity types as shown in Figure 1c& 1d). While eliciting attributions, the models hallucinates the least for named entity persons (`PERSON`), locations and nationalities, religions or

---

[2]We extract named entities using the `en-core-web-trf` model from Spacy (https://spacy.io/).

| « Real Madrid **CF** » « **Real Madrid CF** » Founded on 6 March 1902 as Madrid Football Club, the club has traditionally worn a white home kit since inception. The **honorific title real** is Spanish for **"royal"** and was bestowed to the club by King Alfonso XIII in 1920 together with the royal crown in the emblem. The team has played its home matches in the 81,044-capacity Santiago Bernabéu Stadium in downtown Madrid since 1947. Unlike most European sporting entities, Real Madrid's members (socios) have owned and operated the club throughout its history. | Real Madrid **Club de Fútbol (Spanish pronunciation: [re'al ma'ðrið 'klu$\beta$ ðe 'fuð$\beta$ol] (listen), Royal Madrid Football Club), commonly referred to as Real Madrid, is a Spanish professional football club based in Madrid.** Founded on 6 March 1902 as Madrid Football Club, the club has traditionally worn a white home kit since inception. The **word Real** is Spanish for **Royal** and was bestowed to the club by King Alfonso XIII in 1920 together with the royal crown in the emblem. The team has played its home matches in the 81,044-capacity Santiago Bernabéu Stadium in downtown Madrid since 1947. Unlike most European sporting entities, Real Madrid's members (socios) have owned and operated the club throughout its history. |
|---|---|

Figure 2: Example showing the *difference* (bolded text) between the retrieved passage (on left) vs. the attribution from our `AttributeLLaMA`-65B model (on right), showcasing a high-level of memorization. We found out that this difference is even smaller if we use a latest Real Madrid Wikipedia page instead of the retrieved passage from the old Wikipedia dump.

political groups (`NORP`). They appears to hallucinate the worst in date (`DATE`), time of the day (`TIME`), monetary values and units (`MONEY`) and other numerical measurements (`QUANTITY`).

## 6  DISCUSSION & CONCLUSION

In this work, we introduce `AttributeLLaMA`, a model based on the LLaMA (Touvron et al., 2023) family of models, which is aligned to provide evidences or attributions to the user query. In order to ensure the evidences are honest, i.e. derived from the training corpus of the base LLaMA models, we investigate minimal supervision methods (Zhou et al., 2023). We find that it is possible to finetune LLaMA family of models with only 100 high quality expert annotated data containing question, answer and attribution triplets, in order for the model to generalize and provide attributions to arbitrary user query. We showcase the usefulness of these attributions by improving downstream task results by feeding the model its own generated attributions. We further analyze the generated attributions and discover that they exhibit strong memorization effects from the underlying training data.

Our research focuses on enhancing the interpretability and reliability of LLaMA models. We propose that generating attributions can lead to a significant improvement in interpretability and reliability, even with minimal supervision. This approach is particularly relevant for modern LLMs, which are often prone to memorization (Carlini et al., 2021; 2022).

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

# A    APPENDIX

Table 6: Sample generations of attributions by the `AttributeLLaMA`-70B (Human) on various downstream datasets.

| Dataset | | Generation |
|---|---|---|
| | Instruction | Answer the following question by using the Supporting Document. If supporting document is not provided, generate one and use it to provide the answer. Provide the answer in a few words. |
| NQ | Question | What is the significance of the sahara desert? |
| | Answer | Largest hot desert in the world |
| | Evidence | The Sahara is the largest hot desert and the third largest desert in the world after Antarctica and the Arctic. Its area of 9,200,000 square kilometres (3,600,000 sq mi) is comparable to the area of the United States. The name Śaharaís derived from a dialectal Arabic word for "desert", ṣaḥra. The desert comprises much of North Africa, excluding the fertile region on the Mediterranean Sea coast, the Atlas Mountains of the Maghreb, and the Nile Valley in Egypt and Sudan. It stretches from the Red Sea in the east and the Mediterranean in the north to the Atlantic Ocean in the west, where the landscape gradually changes from desert to coastal plains. To the south, it is bounded by the Sahel, a belt of semi-arid tropical savanna around the Niger River valley and the Sudan Region of Sub-Saharan Africa. The Sahara can be divided into several regions including: the western Sahara, the central Ahaggar Mountains, the Tibesti Mountains, the Aïr Mountains, the Ténéré desert, and the Libyan Desert. |
| | Closest Wiki Match | http://en.wikipedia.org/wiki/SaharaSahara2 |
| | RougeL Score (P/R/F) | 1.0, 0.698, 0.822 |
| WebQ | Prompt | What honor did Agatha Christie receive in 1971? |
| | Answer | Dame Commander of the Order of the British Empire |
| | Evidence | Dame Agatha Mary Clarissa Christie, Lady Mallowan, DBE (née Miller; 15 September 1890 – 12 January 1976) was an English writer. She is known for her 66 detective novels and 14 short story collections, particularly those revolving around her fictional detectives Hercule Poirot and Miss Marple. Christie also wrote the world's longest-running play, a murder mystery, The Mousetrap, and, under the pen name Mary Westmacott, six romances. In 1971 she was appointed a Dame Commander of the Order of the British Empire (DBE) for her contribution to literature. |
| | Closest Wiki match | http://en.wikipedia.org/wiki/Agatha_Christie#Agatha_Christie0 |
| | RougeL Score (P/R/F) | 0.6,0.76,0.673 |
| TriviaQA | Prompt | Where was the fiddler in the musical's title? |
| | Answer | Anatevka |
| | Evidence | Fiddler on the Roof is a musical with music by Jerry Bock, lyrics by Sheldon Harnick, and book by Joseph Stein, set in Tsarist Russia in 1905. It is based on Tevye and his Daughters (or Tevye the Milkman and other tales) by Sholem Aleichem. The story centers on Tevye, the father of five daughters, and his attempts to maintain his Jewish religious and cultural traditions as outside influences encroach upon the family's lives. He must cope both with the strong-willed actions of his three older daughters, who wish to marry for love – each one's choice of a husband moves further away from the customs of his faith – and with the edict of the Tsar that evicts the Jews from their village. |
| | Closest Wiki match | http://en.wikipedia.org/wiki/Fiddler_on_the_Roof#Fiddler_on_the_Roof0 |
| | RougeL Score (P/R/F) | 0.75, 0.792, 0.77 |
| MMLU | Prompt | A question-and-answer dialogue in which propositions are methodically scrutinized to uncover the truth is known as _.
A. an argument
B. the Socratic method
C. the Socratic jest
D. a debate |
| | Answer | B. the Socratic method |
| | Evidence | The Socratic method is a form of cooperative argumentative dialogue between individuals, based on asking and answering questions to stimulate critical thinking and to draw out ideas and underlying presuppositions. It is a dialectical method, that is, a method of inquiry and debate between two or more people holding different points of view. The Socratic method is named after the classical Greek philosopher Socrates, and is a type of pedagogy in which a series of questions is asked not only to draw individual answers, but also to encourage fundamental insight into the issue at hand. |
| | Closest Wiki match | http://en.wikipedia.org/wiki/Socratic_method |
| | RougeL Score (P/R/F) | 0.52, 0.473, 0.497 |

Table 7: Attribution examples from expert-annotated data.

---

Question: Who sings the song 'Riptide'?
Evidence: 'Riptide' is a song by Australian singer-songwriter Vance Joy. It was first released as a track on his debut EP God Loves You When You're Dancing (2013), serving as its second single, and is also featured on his debut studio album Dream Your Life Away (2014). The song was written by Joy, who also produced it with drummer Edwin White. The upbeat song has been lyrically described as a 'coming of age love story' and is known for its metaphors and pop culture references.
Answer: Vance Joy

Question: What was the world's first high level programming language 1957?
Evidence: Fortran was originally developed by IBM in the 1950s for scientific and engineering applications, and subsequently came to dominate scientific computing. It gave computer users the first accessible "high-level" language and enabled computers to optimize commands 20 times more efficiently. It has been in use for over six decades in computationally intensive areas. It is a popular language for high-performance computing and is used for programs that benchmark and rank the world's fastest supercomputers.
Answer: Fortran

Question: Which Dutch artist painted 'Girl with a Pearl Earring'?
Evidence: Girl With A Pearl Earring is an oil painting by Dutch Golden Age painter Johannes Vermeer, dated c. 1665. Going by various names over the centuries, it became known by its present title towards the end of the 20th century after the earring worn by the girl portrayed there. The work has been in the collection of the Mauritshuis in The Hague since 1902 and has been the subject of various literary and cinematic treatments.
Answer: Vermeer

Question: Where did Christopher Columbus sail his ship when he discovered America?
Evidence: San Salvador Island, previously Watling's Island, is an island and district of the Bahamas, famed for possibly being the location of Christopher Columbus's first sighting of the Americas on 12 October 1492 during his first voyage. This historical importance, the island's tropical beaches, and its proximity to the United States have made tourism central to the local economy.
Answer: San Salvador

---

