# OpenReview forum: "Eliciting Attributions from LLMs with Minimal Supervision"
_ICLR.cc/2024/Conference — Submitted to ICLR 2024_

### Official Review · Reviewer_iFFD · 2023-10-28

**Soundness:** 3 good
**Presentation:** 3 good
**Contribution:** 2 fair
**Rating:** 3
**Confidence:** 3

**Summary:**

This paper tackles the fundamental challenge of understanding how Large Language Models (LLMs) encode and retrieve information, aiming to enhance their interpretability and reliability. To address this, the authors introduce AttributeLLaMA, a model extension built upon the LLaMA family of models. The primary objective is to enable LLMs to attribute their generated responses to specific sources of information. Remarkably, this capability is achieved through the fine-tuning of AttributeLLaMA using a mere 100 expert-annotated attribution examples. The results of their experimental studies are compelling, showing that attributions significantly boost LLM performance across a diverse range of downstream tasks, including MMLU and StrategyQA, with improvements exceeding 3.5% and 4.0%, respectively, when relevant attributions are incorporated. In addition to these performance enhancements, their analyses shed light on the exceptional memorization capacities of LLMs, particularly concerning Wikipedia documents included in their training data. This research underscores the potential of generating attributions to enhance the reliability and interpretability of LLMs, even with minimal supervision. This approach holds particular relevance for modern LLMs, which are susceptible to issues of memorization, as highlighted in the study.

**Strengths:**

They employ fine-tuning of LLMs to generate attributions, and the experimental results demonstrate that by incorporating these attributions, the model achieves significantly enhanced performance.

**Weaknesses:**

* The paper solely compares its approach with a baseline that lacks any provided attributes, overlooking potential comparisons with alternative methods, such as a chain-of-thought approach that generates evidence before responding to questions.
* A comparison between the attributes generated by their model and those from a baseline model without fine-tuning would provide a valuable benchmark for evaluating the effectiveness of their approach, which is currently absent from the study.
* The absence of clear advancements in the results when applied to larger models raises concerns about the practical utility and scalability of their model, prompting questions regarding its broader applicability.

**Questions:**

N/A

---

### Official Review · Reviewer_gxgc · 2023-10-31

**Soundness:** 3 good
**Presentation:** 3 good
**Contribution:** 2 fair
**Rating:** 3
**Confidence:** 4

**Summary:**

The authors present AttributeLLaMA, a novel method for improving the interpretability of Large Language Models (LLMs) by enabling them to provide attributions for their responses. The authors leverage the LLaMA model, fine-tuning it with 100 expert-annotated attribution examples to create AttributeLLaMA. Evaluations on diverse downstream tasks, such as MMLU and StrategyQA, show significant performance improvements when leveraging these attributions.

Furthermore, the authors conduct an analysis, revealing that attributions obtained from AttributeLLaMA highlight the model's remarkable memorization capabilities. The authors also explore the impact of the number of attention heads on model parallel scaling and provide strong scaling results for the 1.2 billion parameter model.

**Strengths:**

This work is well-motivated. The ability of a model to provide attributions for its decisions can enhance trust in the model's predictions and improve debugging capabilities. The AttributeLLaMA is interesting and efficient given the small amount of data required for effective training.

The paper provides comprehensive experiments and analyses on multiple datasets and tasks, demonstrating the effectiveness of the proposed method. The results show significant improvements over baseline models.

**Weaknesses:**

My biggest concern about this work is that the paper does not provide a clear comparison with other existing models (instruction-following baselines) and methods for interpretability in LLMs. This makes it very difficult to fully understand the novelty and effectiveness of the proposed approach.

The data selected for fine-tuning is close to the downstream task,  whether the improvements are brought by attributions is unclear. While the authors discuss the memorization capabilities of the model which is interesting, they do not delve deeply into the potential implications of this, such as the risk of overfitting or the possibility of the model memorizing sensitive information.

In conclusion, whilst the simplicity of the work is appreciated, the paper lacks the depth of insightful conclusions and novel findings. This renders the study somewhat overly straightforward, akin to the style of a technical blog rather than academic research.  We still do not know the principles of how to construct more high quality attributions to train LLMs after reading this paper.

**Questions:**

1. Could the authors compare with other existing methods for eliciting attributions or improving interpretability in LLMs? This would help to better position the work in the field and understand its novelty. Since this work uses the Dolly dataset, a comparison with other instructions-following models is needed at least.

2. The authors might consider exploring the impact of varying the number of expert-annotated examples used for fine-tuning. This could provide insights into the trade-off between the amount of supervision and the performance of the model.

3. The authors could provide more details on the metrics about how they measure the quality of attributions that would guide future better demonstrate how the attributions improve the model.

---

### Official Review · Reviewer_ZtX9 · 2023-11-01

**Soundness:** 2 fair
**Presentation:** 3 good
**Contribution:** 2 fair
**Rating:** 3
**Confidence:** 4

**Summary:**

This work proposes a model AttributeLLaMa, which can produce evidence or context for QA tasks. This work proposes to use minimum supervision to elicit the model's ability on generating relevant evidence. Then the generated evidence will be used as the context for a QA model to answer the question. This work conducts a variety experiments on various QA datasets, and experimental results show that the generated evidence can enhance the performance of the QA model.

**Strengths:**

1. The idea of adopting minimum supervision to elicit the model ability on generating evidence is interesting and seems to be effective according to the performance reported in the work.

2. This work conduct experiments on various QA datasets of different types, which can support the claim of the proposed method.

3. The discussion of memorization and hallucination in Section 5 is encouraged and desired.

**Weaknesses:**

1. I do not really get the necessity of separating the two tasks - evidence generation and QA - apart into two steps. As claimed in the paper, only 100 examples can elicit the ability of the model to generate evidence. Then why not use the same model to perform evidence generation and QA at the same time, by just calling the model once? Now the two-step process is more like we are using LLMs as an external knowledge base, which can be used for provide knowledge for another LLM for performing QA tasks. From this perspective, there are many existing literatures that have explored such a problem.

2. The claim of "minimum supervision" seems to be oversimplified. How does the number 100 come from? Why it is the minimum supervision? Can this supervision be replaced by other machine generated (evidence, question-answer) pair? These questions need to be resolved in the paper.

3. In general I think the contribution of the paper is not strong enough. Although the proposed framework seems effective, it considers a normal way that how we can utilize LLMs as knowledge source.

**Questions:**

Please refer to the questions in the weakness.

---

### Official Review · Reviewer_QHns · 2023-11-09

**Soundness:** 3 good
**Presentation:** 3 good
**Contribution:** 3 good
**Rating:** 5
**Confidence:** 3

**Summary:**

In this paper, the authors collected 100 high-quality question-answering examples and attribution annotated by experts. They fine-tuned LLaMa on this data to enable the model to output evidence (attribution) for questions, based on the knowledge learned during the model pre-training. The authors conducted a series of evaluation experiments on various QA benchmarks and observed that LLaMa when equipped with generated attributions, outperforms the baseline model without it.

**Strengths:**

1. The small, expertly annotated dataset is of high quality, which offers resources for future research works.
2. The approach is impressive and effective, demonstrating that fine-tuning with data of only 100 annotated attributions is sufficient to enable LLaMa to recall knowledge from its pre-training phase.
3. The analysis is thorough, evaluating improvements on QA benchmarks as well as examining the reasoning, memorization, and hallucination phenomena.

**Weaknesses:**

1. My primary concern is the lack of novelty and innovation: The method of prompting language models to generate evidence as attribution is not new, and the difference between this work and GenRead has not been discussed. The key contribution appears to be the annotation of a small dataset and the following fine-tuning on one single LLM, which may not be generalizable. The dataset's high quality could be tailored to this particular model, limiting the broader applicability of the findings.
2. The second concern is related to the reproducibility and generalizability of the work. There is a lack of detailed information on the data selection/annotation procedures, as well as a comprehensive analysis of why 100 examples are sufficient. It remains unclear whether selecting a random set of 100 QA pairs from another benchmark and asking for user annotations would also successfully elicit attributions from a large language model after fine-tuning.
3. The prompt design is confusing: The authors compare LLaMa with LLaMa+AttributeLLaMa. However, after fine-tuning, AttributeLLaMa is capable of generating not just the evidence but also the answer, following the designed instructions. It is unclear why the direct outputs (answers) were not able to be directly used and compared (i.e., LLaMa vs. AttributeLLaMa).

**Questions:**

1. Could you provide more details on the annotated data to prove its quality? For example, any important statistics and details about the annotation procedure? Did each expert label 50 examples, so 100 examples in total? Or did they annotate 100 examples together and deal with the discrepancy? Can you provide insights into why these 100 examples are sufficient?
2. Could you clarify what GenRead-175B refers to? Is it GenRead with InstructGPT-175B backbone? What is the comparison on QA performance if compared with more recent language models, such as LLaMa2, GPT-4, or ChatGPT; and those models with attributions (e.g. ask them to answer questions with attributions by CoT)?

---

### Meta-Review · Area_Chair_5hNG · 2023-12-20

**Metareview:**

The authors proposes AttributeLLaMA, a method to examine attributions for LLM's generation. The authors leverage the LLaMA model, fine-tuning it with 100 expert-annotated attribution examples to create AttributeLLaMA. Evaluations on diverse downstream tasks, such as MMLU and StrategyQA, show significant performance improvements when leveraging these attributions. Reviewers have major concerns about comparison with existing methods to interpret LLMs. It is unclear about the main insight from this paper other than instruction fine-tuning.

**Justification For Why Not Higher Score:**

Clear weakness in the paper and there is no response from authors.

**Justification For Why Not Lower Score:**

N/A

---

### Decision · Program_Chairs · 2024-01-16

Reject